# The Association of Obstructive Sleep Apnea Syndrome and Accident Risk in Heavy Equipment Operators

**DOI:** 10.3390/medicina55090599

**Published:** 2019-09-17

**Authors:** Hakan Celikhisar, Gulay Dasdemir Ilkhan

**Affiliations:** 1Department of Chest Deseases, İzmir Metropolitan Municipality Hospital, İzmir 35110, Turkey; 2Department of Chest Diseases, Okmeydanı Training and Research Hospital, Istanbul 34384, Turkey; gulaydasdemir@gmail.com

**Keywords:** polysomnography, heavy equipment operators, STOP BANG Questionnaire, Obstructive Sleep Apnea Syndrome

## Abstract

*Background and Objectives*: Obstructive sleep apnea syndrome (OSAS) is the most frequent sleep disorder, characterized by the repeated collapse of the upper respiratory tract during sleep. In this study, we aimed to determine the prevalence of OSAS in heavy equipment operators and to determine the relationship between the work accidents that these operators were involved in and the OSAS symptoms and severity. In doing this, we aimed to emphasize the association of OSAS, which is a treatable disease, and these accidents, which cause loss of manpower, financial hampering, and even death. *Materials and Methods*: STOP BANG questionnaire was provided to 965 heavy equipment operators and polysomnography (PSG) was performed, in Izmir Esrefpasa Municipality Hospital, to the operators at high risk for OSAS. Demographic data, health status, and accidents of these operators were recorded. *Results*: All operators who participated in the study were male. The ages of the cases ranged from 35 to 58 and the mean age was 45.07 ± 5.54 years. The mean STOP BANG questionnaire results were 4.36 ± 3.82. In total, 142 operators were identified with high risk for OSAS and PSG could be performed on 110 of these 142 operators. According to the PSG results of the operators, 41 (37.3%) patients had normal findings, while 35 (31.8%) had mild, 20 (18.2%) had moderate, and 14 (12.7%) had severe OSAS. Among those 110 patients, 71 (64.5%) of the cases had no history of any accidents, 25 (22.8%) were almost involved in an accident due to sleepiness, and 14 (12.7%) were actually involved in an accident. There was a statistically significant relationship between the accident rate and OSAS severity (*p*: 0.009). *Conclusion*: Based on the data acquired in the present study, a positive correlation was determined between the accident statuses of drivers with OSAS severity. We want to attract attention to the necessity of evaluating the OSAS symptoms in professional heavy equipment operators during the certification period and at various intervals afterwards, and to carry out OSAS evaluations by PSG for those having a certain risk.

## 1. Introduction

Sleep is the temporary, partial, periodic, and reversible loss of the communication of the organism with the environment, which is an indispensable factor for a healthy life [1]. Obstructive sleep apnea syndrome (OSAS) is characterized by the repeated collapse of the upper respiratory tract during sleep, causing nocturnal hypoxemia and interrupted sleep [1]. It is the most frequent sleep disorder. OSAS prevalence has been determined as 3%–7% for males and 2%–5% for females all over the world [2].

The most common night symptom of OSAS is snoring, while the day symptom is excessive sleepiness [3,4]. The most important risk factors are indicated as male gender, advanced age, neck circumference, and obesity [5].

Various questionnaires are used for identifying risky groups and Berlin questionnaire is one of these arranged for community screenings. There are a total of 10 questions in three categories. Positive results in two or more categories indicate that the participant is carrying high risk [6]. STOP-BANG is another questionnaire defined to have high sensitivity to predict OSAS [7]. Polysomnography (PSG) is the gold standard in OSAS diagnosis and treatment management [5]. OSAS has been classified into three different classes according to apnea hypopnea index (AHI) as mild OSAS (AHI = 5–15), moderate OSAS (AHI = 15–30), and severe OSAS (AHI > 30) in accordance with the American Academy of Sleep Medicine Criteria [5]. Continuous positive air pressure (CPAP) is the standard treatment for OSAS [8,9].

Even though the nighttime symptoms of OSAS are generally ignored by the patient, its daytime symptoms are quite striking. Daytime excessive sleepiness may be so severe that it may affect work performance, prevent driving a vehicle carefully, and increase the accident risks [3]. For that reason, patients under high risk for OSAS should be diagnosed and treated. In this study, we aimed to determine the prevalence of OSAS in heavy equipment operators and to determine the relationship between the work accidents that these operators were involved in and the OSAS symptoms and severity. In this way, we aimed to emphasize the association of OSAS, which is a treatable disease, and these work accidents causing loss of manpower, financial hampering, and even death, which may be preventable.

## 2. Patients and Methods

The present study was planned as a prospective study at the Esrefpasa Municipality Hospital after the ethical approval was obtained from the Metropolitan Municipality (Number 54022451-050.05-04- from 1 February 2017). STOP BANG questionnaire was applied to 965 heavy equipment operators and PSG was performed in Izmir Esrefpasa Municipality Hospital to the operators at high risk for OSAS. The machines used by the operators include dumpers, hydraulic backhoes, crawled dozers, graders, wheeled front-end loaders, wheeled vibrating rollers, hydraulic breakers, drills, bucked wheel excavators, and backhoe loaders. Signed informed consent forms were obtained from all participants.

For the determination of STOP-BANG scores, snoring, daytime sleepiness (tiredness), observed apnea, high blood pressure (antihypertensive drug use), body mass index (positive if BMI > 35), age (positive if > 50 years), neck circumference (positive if > 40 cm.), and gender (male gender positive) were recorded. If participants chose the answer ‘yes’ for 3 of 8 questions; they were accepted as having high risk for OSAS.

Between February 2017 and March 2019, all of the operators examined in our hospital were male. Demographic characteristics such as age, weight, height, body mass index, neck circumference, waist/hip ratio, alcohol and cigarette smoking, and medical history were recorded. At the same time, the municipal official records of the accidents were questioned and recorded.

All patients included in our study were monitored all night by a trained sleep technician using a PSG device at our sleep center. At least 6 h of PSG recordings were acquired. Sleep staging and respiratory- and movement-scoring were done according to the American Academy of Sleep Medicine manual (version 2.0) by a sleep technician [10]. Apnea hypopnea index (AHI) was defined as the number of apneas and hypopneas per hour of sleep. Apnea was defined as the drop of airflow ≥90% of baseline for at least 10 s and hypopnea as a decrease in airflow of at least 30% for at least 10 s with oxygen desaturation of more than 4% from baseline. The severity of OSA was determined by the AHI as mild if AHI is between 5 and 15, as moderate if AHI is between 15 and 30, and severe if AHI is greater than 30.

## 3. Statistical Analyses 

IBM SPSS Statistics v. 22 (IBM Corp., Armonk, NY, USA) software was used for statistical analyses. Shapiro–Wilks test was used for evaluating the accordance of the parameters with normal distribution. In addition to descriptive statistical methods (mean, standard deviation, frequency), one-way ANOVA test was performed for the comparison of quantitative data, as well as Tukey HDS test and Tamhane’s T2 test for the intergroup comparison of parameters with normal distribution and the determination of the group that causes the difference. Kruskal–Wallis test was used for carrying out intergroup comparisons of parameters without normal distribution. Whereas Chi Square test and Fisher–Freeman–Halton tests were performed for comparing qualitative data. Regression analysis was performed to determine the factors affecting the accident risk. Level of significance was evaluated as *p* < 0.05.

## 4. Results 

The study was carried out between February 2017 and March 2019. In total, 142 operators were identified with high risk for OSAS and PSG could be performed on 110 of these 142 operators. All operators were male and their ages ranged from 35 to 58 years. The mean age of the operators was 45.07 ± 5.54 years. According to the PSG results of the study, 41 (37.3%) of the operators had normal findings, 35 (31.8%) had mild, 20 (18.2%) had moderate, and 14 (12.7%) had severe OSAS. The distribution of the general characteristics of the operators who participated in the study is shown in Table 1.

Of the cases, 87 (79.1%) did not have an accident, while 18 (16.4%) almost had an accident, and 5 (4.5%) had an accident. While 41 (37.3%) of the patients had normal PSG results, 35 (31.8%) had mild, 20 (18.2%) had moderate, and 14 (12.7%) had severe OSAS. The distribution of OSAS classification and accident status is summarized in Table 2. There was a statistically significant relationship between the accident rate and OSAS severity (*p*: 0.009).

The present results indicate that there was a statistically significant difference between accident histories with regard to apnea prevalence (*p*: 0.009). Apnea prevalence in those who had not been involved in any accident (37.2%) was observed to be lower at a statistically significant level in comparison to those who were almost involved in an accident due to sleepiness (64.8%) and those who were involved in an accident (61%) (*p*_1_: 0.019; *p*_2_: 0.047, respectively). No statistically significant difference with regard to apnea prevalence was observed between those who were almost involved in an accident due to sleepiness and those who had been involved in an accident (*p* > 0.05). A statistically significant difference was observed between accident histories with regard to snoring + apnea prevalence (*p*: 0.004). The snoring + apnea prevalence in those who had not been involved in any accident (35.8%) was observed to be lower at a statistically significant level in comparison to those who were almost involved in an accident (63.1%) and those who had been involved in an accident (62%) (*p*_1_: 0.011; *p*_2_: 0.031, respectively).

Table 3 presents the assessment of age, BMI, neck circumference, waist–hip ratio, and Epworth score among the OSAS classification groups.

According to the results of the present study, there was a statistically significant difference between the OSAS groups determined by PSG, regarding the BMI values and neck circumference (*p*: 0.001). The patients without OSAS had statistically significantly lower BMI values than those with moderate and severe OSAS (*p*_1_: 0.032; *p*_2_: 0.000, respectively). The BMI values of those with severe OSAS were determined to be higher at a statistically significant level in comparison with the BMI values of those with mild or moderate OSAS (*p*_1_: 0.004; *p*_2_: 0.017, respectively). The neck circumference values of those without OSAS were determined to be lower at a statistically significant level in comparison with that of the patients with moderate or severe OSAS (*p*_1_: 0.019; *p*_2_: 0.001, respectively). Neck circumference values of those with severe OSAS were determined to be higher at a statistically significant level in comparison with the mild OSAS group (*p*: 0.001). The waist–hip ratio of individuals with mild OSAS was determined to be lower at a statistically significant level in comparison with the waist–hip ratio of individuals with moderate or severe OSAS (*p*_1_: 0.046; *p*_2_: 0.004, respectively). Epworth score of the severe OSAS group was significantly higher than that of the healthy cases (*p*: 0.001).

There was a statistically significant difference between the OSAS groups with regard to accident status (*p*: 0.001;). The ratio of being almost involved in an accident due to sleepiness (0%) for drivers without OSAS was determined to be lower at a statistically significant level in comparison to drivers with mild (15.4%), moderate (28.6%), or severe (39.6%) OSAS (*p*_1_: 0.026; *p*_2_: 0.001; *p*_3_: 0.001, respectively). The ratio of being involved in an accident for drivers with severe OSAS (34%) was determined to be higher at a statistically significant level in comparison to those of drivers with mild (0%) or moderate (5.7%) OSAS (p:0.001). No statistically significant difference could be determined between the drivers with mild and moderate OSAS with regard to the ratios of being involved in an accident (*p* > 0.05).

In Table 4, the results of regression analysis performed to determine the factors increasing accident risk are summarized. Regarding these findings, only AHI had significant effects on accident risk.

## 5. Discussion

Daytime sleepiness and loss of concentration is a common cause of accidents [11]. Short-term and poor quality sleep causes excessive daytime sleepiness, which increases the risk of accidents. Sleep disturbances such as obstructive sleep apnea syndrome are common causes of excessive sleepiness. In this study, in heavy equipment operators we determined a significant increase in accident risk with an increase in OSAS severity and in a regression analysis, there was a strong relationship between the AHI score and accident risk.

The studies on this subject are generally performed on long-distance drivers or taxi drivers, and the number of studies with the heavy equipment operators is limited. The majority of these studies are survey-oriented and the results obtained from the surveys are based on subjective data. Various surveys conducted with truck drivers showed a positive correlation between daytime excessive sleepiness and accidents [12,13,14]. In our study, we found that daytime sleepiness in heavy equipment operators was not a significant risk factor for accidents. However, this result may be related to the fact that the work was planned specifically for the heavy equipment operators and the maximum speed of the machines they were driving was limited at a speed of 20 km/h. However, in the case of heavy equipment operators, a distraction may cause serious accidents that can result in loss of life and property [15].

According to the results of STOP BANG survey, operators who were found to be at high risk for OSAS were evaluated with the results of PSG, performed in our sleep laboratory. Our study is one of the studies with a large sample group, which was performed with the participation of professional operators and evaluated the relationship between the accidents and PSG results. On the one hand, working with objective data will give more accurate results, on the other hand, the difficulty of doing PSG for each participant may be the obvious reason for not giving up surveys. Subjective symptoms such as snoring and apnea, which are questioned in terms of OSAS, are the symptoms they cannot detect on their own. On the other hand, financial anxiety may result in bias and the surveys may not reflect the realities. Results with similar concerns may be deceptive during job applications or health checks. Therefore, OSAS symptom questioning alone may be inadequate, and delays in diagnosis and treatment of OSAS cases may cause severe issues for the patients as well as the society due to the accidents [16,17]. With this fact, in some parts of the European Union countries, driver’s licenses are not given to OSAS patients because they do not have healthy driver qualifications [18,19].

The STOP BANG questionnaire was used to determine daytime sleepiness. In our study, the STOP BANG questionnaire score was below three in the majority of patients, including those describing daytime sleepiness (EDS). Regarding this data, the STOP BANG questionnaire completed by the operators may be suggested as inadequate and unsuitable for determining the EDS. We also did not determine an association between STOP BANG score and accident risk in regression analysis. On the other hand, the work and material concerns of our patients may make the questionnaire-based survey results illusory.

When compared to normal healthy individuals, accidents with OSAS operators were reported to be seen seven times higher [20,21]. As the apnea hypopnea index (AHI) increased in the study of Young et al., the risk of accident was found to be increased, in accordance with our study [22]. In another study by Teran-Santos et al., a significant relationship was found between the risk of accident and presence of OSAS. According to the results of the same study, when the operators with OSAS were evaluated with the increase in severity of the disease, the risk of accidents also increased, in parallel with our study. Accident rates in our study were determined as 28.5% in severe OSAS, 5.0% in moderate OSAS, and 0% in mild OSAS patients. Statistically, these rates show a positive correlation between the severity of OSAS and accident frequency. On the other hand, according to the results of STOP BANG survey, PSG was performed on a high-risk group and some of them did not have OSAS. No accident history was found in healthy group. Regarding the demographic data in our study; body mass index (BMI), neck circumference, and waist–hip ratio did not show any association with the accident risk. However, in the study of Amra et al., increased neck circumference was determined as a factor increasing the risk of accident [13].

In the light of all these data, it seems that the questioning of OSAS symptoms alone will not be enough to predict the risk of accidents. However, there are many studies demonstrating the relationship between the severity of OSAS and accident risk. Regarding the increased morbidity and mortality due to the accidents, it is a necessity to inquire about the OSAS symptoms in the professional drivers, such as heavy equipment operators, during the certification phase and to perform polysomnography on the drivers with high risk. It is concluded that in order to prevent the morbidity and mortalities, in the groups involved in the field of occupational risk, routine PSG studies may be required. For this purpose, it is necessary to question the number of sleep centers and to determine the OSAS symptoms periodically during the certification and after, and to increase the screening.

There are some limitations of this study that should be mentioned. The STOP BANG questionnaire is based on the patient responses, and this may carry some bias that the vehicle operators may not respond correctly; since they may be afraid of losing their job. Second, this is the report of single center results and larger, prospective studies investigating the treatment responses in this group of patients are warranted.

## 6. Conclusions

We want to attract attention to the necessity of evaluating the OSAS symptoms of professional heavy equipment operators during the certification period and at various intervals afterwards and to carry out OSAS evaluations by PSG for those with a certain risk.

## Figures and Tables

**Table 1 medicina-55-00599-t001:** Distribution of general characteristics.

	Min-Max	Mean ± SD
**Age**	35–58	42.07 ± 5.54
**BMI * (kg/m^2^)**	21.89–42.14	31.89 ± 3.67
**Neck circumference (cm)**	34–52	42.32 ± 2.65
**Waist to Hip ratio**	0.81–1.13	0.96 ± 0.04
**STOP BANG score**	0–8	3.47 ± 3.73
**Number of cigarette packs smoked annually**	1–44	18.65 ± 7.89
**Smoking**		
***Never smoked***	34	30.90%
Quit smoking	20	18.50%
Current smoker	56	50.60%
**Alcohol**		
***Not*** drinking	81	73.50%
Drinking	29	26.50%

*: Body Mass Index.

**Table 2 medicina-55-00599-t002:** Distribution of obstructive sleep apnea syndrome (OSAS) classification and accident status.

	Normal (n: 41)	Mild OSAS (n: 35)	Moderate OSAS (n: 20)	Severe OSAS (n: 14)
**No accident history**	41 (100%)	29 (82.8%)	13 (65.0%)	4 (28.5%)
**Almost involved in an accident**	0	6 (17.2%)	6 (30.0%)	6 (43.0%)
**Involved in an accident**	0	0	1 (5.0%)	4 (28.5%)

**Table 3 medicina-55-00599-t003:** Evaluation of age, BMI, neck circumference, waist-hip ratio and STOP BANG, smoking and alcohol use states among OSAS classification groups.

	OSAS * Classification				*p*
	Normal (n: 41)	Mild OSAS (n: 35)	Moderate OSAS (n: 20)	Severe OSAS (n: 14)	
	Mean ± SD	Mean ± SD	Mean ± SD	Mean ± SD	
**Age (years)**	42.31 ± 5.8	41.39 ± 5.62	42.32 ± 6.08	42.34 ± 4.89	**0.866**
**BMI ** (kg/m^2^)**	28.79 ± 2.7	30.98 ± 4.56	31.98 ± 3.31	34.01 ± 2.89	**0.001 ***
**Neck circumference (cm)**	40.99 ± 2.2	40.51 ± 3.22	41.45 ± 2.54	44.78 ± 2.1	**0.001 ***
**Waist–Hip ratio**	0.94 ± 0.04	0.94 ± 0.06	0.97 ± 0.04	0.98 ± 0.65	**0.002 ***
**STOP BANG score**	3.91 ± 2.4	4.69 ± 3.69	4.89 ± 4.59	4.37 ± 3.89	**0.952**
**Epworth score**	4.11 ± 2.09	4.31 ± 2.21	4.71 ± 2.21	4.89 ± 2.04	**0.012**

*: Obstructive sleep apnea syndrome, **: Body mass index.

**Table 4 medicina-55-00599-t004:** Regression analysis performed to determine the factors increasing accident risk.

	t	*p*
**BMI**	−1.886	0.071
**AHI**	7.960	0.001
**Epworth score**	1.572	0.118
**Waist/hip ratio**	−0.291	0.772
**Neck**	1.107	0.270
**STOP BANG score**	−1.486	0.112

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
