# Peer review of "The Association of Obstructive Sleep Apnea Syndrome and Accident Risk in Heavy Equipment Operators"

_medicina, 2019, doi:10.3390/medicina55090599_

Round 1

Reviewer 1 Report

Thank you very much for possibility to review this manuscript!

The paper is well written and easy to read.

The manuscript is dedicated to study the Obstructive sleep apnea syndrome frequency and 2 accident risk in heavy equipment operators. Methods and results are consistent. The discussion is short and simple, and in my opinion must be revised.

Although the present data are intriguing, there are several issues that should be addressed:

The manuscript could benefit from being proof read/edited by an English speaker to clarify multiple grammatical and linguistic errors. Please add introduction in abstract. Aims in the abstract must be reformulated Introduction must be revised: give more data on your topic not only general and very well known things Discussion must be revised: you need to compare more deeply your data with data from literature Statistics is only descriptive: in my opinion you must perform regression analysis for determination of the risk factors for accidents 98-99 Of the cases, 71 (64.8%) did not have an accident, while 25 (22.8%) had an accident and 14 (12.3%) had an accident. There is a mistake in the sentence, there are two groups of patients who had accidents. Please recalculate the percentage, since it is 99.9% Similarly recalculate the percentage for other tables since summ it is often 99.9% In table 2 and 3 please change light to mild!!! Please add data on Epworth!!! There is no any data about if this study was approved by Ethics committee Please add limitations of your study

Author Response

Response to the reviewer 1 Comments:

Point 1: The manuscript could benefit from being proof read/edited by an English speaker to clarify multiple grammatical and linguistic errors.

Response 1: Extensive editing is performed for the English language and grammatical and linguistic errors are clarified.

Point 2: Please add introduction in abstract. Aims in the abstract must be reformulated Introduction must be revised: give more data on your topic not only general and very well known things.

Response 2: The abstract pat is re-written. The “aims” part is added. Introduction is improved

Point 3: Discussion must be revised

Response 3: The discussion and conclusion parts are revised and improved. The results are more clearly presented.

Point 4: Please add data on Epworth!

Response 4: The data on Epworth is added.

Point 5: In table 2 and 3 please change light to mild.

Response 5: The term ‘light’ is changed with ‘mild’ in tables.

Response to the reviewer 2 Comments:

Point 1: Abstract: It is too long with unnecessary verbiage.

Response 1: The abstract is re-written.

Point 2: 1stparagraph is unnecessary and doesn’t contribute to the paper. There are better references to cite for the prevalence of OSA.

Response 2: In the introduction part; revisions were performed as addressed by you. Thank you.

Point 3: The fifth reference should be corrected.

Response 3: The fifth reference is corrected.

Point 4: The method and results are presented in a very confusing manner.

Response 4: The methods and results are more clarified.

Point 5: Focus the discussion on the main points you are trying to make.

Response 5: The discussion part is revised and improved.

Thank you very much for your encouraging comments. We studied on the points you mentioned and tried to correct all points. It will be grateful if you would re-evaluate the manuscript.

Kind regards.

Reviewer 2 Report

General Comments:

            This study by Celikhisar and Ilkhan is a prospective survey of heavy equipment operators in Turkey. The authors found that there was a high prevalence of obstructive sleep apnea (OSA) in this population, and that there was an association between severity of OSA and risk of an accident. They administered the STOP-BANG questionnaire and found no association between either severity of OSA or risk of an accident. These findings are certainly of local and perhaps regional interest, but from a broader perspective would be expected and are not particularly novel. There are also some issues regarding the clarity of the methods as well as the organization of the paper. Those are detailed below under Specific Comments.

Specific Comments:

            Abstract: It is too long with unnecessary verbiage.

            Introduction:

1stparagraph is unnecessary and doesn’t contribute to the paper. There are better references to cite for the prevalence of OSA. Laratta et al doesn’t cite worldwide prevalence rates. I suggest Punjabi NM. The epidemiology of adult obstructive sleep apnea. Proc Am Thorac Soc. 2008 Feb 15;5(2):136-43. doi: 10.1513/pats.200709-155MG. Review. PubMed PMID: 18250205; PubMed Central PMCID: PMC2645248. You should say “gold standard” not ‘golden standard’ Reference 5 is incorrect. It should be

o   Foldvary-Schaefer NR, Waters TE. Sleep-Disordered Breathing. Continuum (Minneap Minn). 2017 Aug;23(4, Sleep Neurology):1093-1116. doi: 10.1212/01.CON.0000522245.13784.f6. Review. PubMed PMID: 28777178.·      American Sleep Disorders Association is incorrect. It should be the American Academy of Sleep Medicine.Methods:

It is unclear how the subjects were recruited. You state “A total of 142 operators were included in the study who have been directed to the sleep laboratory of our hospital after being identified as risky with regard to OSAS symptoms according to the STOP BANG Questionnaire results.” Why were they given the STOP BANG? Was it routine for all operators or was it because these operators had exhibited some symptom of OSAS? The next sentence sentence is unclear. Do you mean to say that the STOP BANG was administered again in addition to checking for antihypertensive drug use, waist/hip etc? Or, are you trying to define the components of the STOP BANG? The description of your polysomnography is inadequate. Reference #1 does not supply the parameters of polysomngraphy described by the American Academy of Sleep Medicine. You should describe the technique especially your definition of hypopnea scoring. The correct reference is to the AASM Scoring Manual. https://aasm.org/clinical-resources/scoring-manual/

Results:

The results are presented in a very confusing manner. Why is the classification of OSA severity presented in text and not in a table? Additionally, there is a problem with the number of subjects. In Table 2, the number of subjects adds up to 162. The methods state that there were 142 subjects. Something is wrong. Much of the results are extraneous to the point you are trying to make which are: Unrecognized OSA is common in this population OSA is associated with accident risk The STOP BANG is a poor instrument to identify accident risk in this population Other anthropometric indices do not appear to have any value THEREFORE SHORTEN AND FOCUS THE RESULTS TO THE ISSUES YOU ARE TRYING TO ADDRESS How was sleepiness identified? The STOP BANG? The STOP BANG is not usually used to do this.

Discussion:

The STOP BANG is not an instrument to determine sleepiness. It is used to identify OSA. Focus the discussion on the main points you are trying to make. Currently it rambles. The anthropometric relationships detract from your discussion of the main findings. It is not surprising that symptoms do not predict accident risk in vehicle operators. It has long been recognized that truck drivers, for example, will not admit that they have symptoms of OSA because this will result in restrictions on their driving licenses, thus impacting their ability to work. This issue needs comment in your paper.

Author Response

(The authors gave the same response as above.)

Round 2

Reviewer 1 Report

I propose to accept at this stage

Author Response

The manuscript was reviewed in line with your reviews and suggestions.

Extensive editing was performed for the English language and grammatical and linguistic errors were clarified.

The revisions were corrected as requested.

                                                                          Best regards.

Reviewer 2 Report

The authors have attempted to address my previous concerns. I no longer have any questions regarding the methods and data analyses.

The English writing, while acceptable, is not optimum. It would be subject to extensive copy editing at a cost to the authors if this paper was accepted in a journal published in an native English speaking country. 

In my opinion, the discussion is still too long, but that is an editorial decision.

Author Response

I've noticed problems with writing in English through your comments. Thank you for that.

We reviewed English and style problems with professional support and corrected our grammatical mistakes.